# A 25-Year Retrospective on Bavaria’s Newborn Screening Programme: Achievements, Challenges and Long-Term Follow-Up

**DOI:** 10.3390/ijns11040114

**Published:** 2025-12-13

**Authors:** Uta Nennstiel, Inken Brockow, Birgit Odenwald, Carola Marzi, Marianne Hanauer, Esther Maier, Wulf Röschinger, Ralph Fingerhut, Bernhard Liebl

**Affiliations:** 1Newborn Screening Centre/State Institute of Health, Bavarian Health and Food Safety Authority, 85764 Oberschleissheim, Germany; birgit.odenwald@lgl.bayern.de (B.O.); carola.marzi@lgl.bayern.de (C.M.); marianne.hanauer@lgl.bayern.de (M.H.); bernhard.liebl@t-online.de (B.L.); 2Newborn Screening Laboratory, Labor Becker MVZ, 81737 Muenchen, Germany; e.maier@labor-becker.de (E.M.); w.roeschinger@labor-becker.de (W.R.); 3Newborn Screening Laboratory, SYNLAB MVZ Weiden, 92637 Weiden, Germany; ralph.fingerhut@synlab.com

**Keywords:** newborn screening, dried bloodspots, tracking, long-term follow-up study

## Abstract

The German federal state of Bavaria implemented newborn screening (NBS) using dried blood spots (DBS) as an integrated public health programme with centralised coordination. The Bavarian NBS Centre collaborates with NBS laboratories, obstetric and paediatric facilities, specialised centres of expertise, and parents. It is responsible for coordination, evaluation, quality assurance, and a long-term follow-up study. In this paper, an analysis of NBS in Bavaria from 1999 to 2023 and a long-term follow-up for the birth cohort until 2013 is presented. Of the 2,854,190 babies screened, 2500 were diagnosed and treated early thanks to NBS. An NBS coverage rate of 99.83% was achieved, with 99.09% of all requested repeat tests completed. Around 87% of infants with time-sensitive conditions underwent a clinical intervention within the first 14 days of life. Systematic tracking enabled all but 54 NBS-positive results to be clarified and 122 newborns to be diagnosed in due time. The results of the long-term follow-up study demonstrate that almost all the children identified through NBS receive ongoing medical care, and that NBS has contributed to the age-appropriate development of most affected children. This 25-year evaluation of NBS in Bavaria shows that near-universal participation in NBS and follow-up of almost all positive NBS results can be achieved through centralised coordination and ongoing cooperation of all those involved.

## 1. Introduction

Newborn screening (NBS) for defined target conditions using dried blood spots (DBS) is widely regarded as one of the most effective measures of population-based secondary prevention [1,2,3,4]. Requirements for an effective NBS include a high coverage of newborns [5,6], comprehensive follow-up of all positive findings, early diagnosis, and prompt medical care for affected children [6,7]. The early diagnosis of NBS conditions is crucial to prevent severe health consequences such as lifelong disabilities or death [8,9,10].

Since the late 1960s, newborns in Germany have been subject to screening for phenylketonuria (PKU), and since the early 1980s, also for congenital hypothyroidism and galactosemia [11]. The advent of tandem mass spectrometry (MS/MS) and other innovative analytical techniques has enabled screening for a broader range of conditions. In the German federal state of Bavaria, a pilot project titled ‘Expanded Newborn Screening in Bavaria’ [12] was conducted between 1999 and 2003 to assess these analytical methods, determine the optimal procedures, and evaluate the integration of additional conditions into NBS based on the screening criteria established by Wilson and Jungner [13]. As part of this pilot project, all infants born in Bavaria were to undergo screening for hypothyroidism, congenital adrenal hyperplasia (CAH), and 20 inborn errors of metabolism (IEMs) within the first 3 days of life. The evaluation of this project revealed that only the 2 endocrinological conditions and 10 of the IEMs met the established NBS criteria [13]. For the remaining conditions, these criteria were not met in terms of either improving health outcomes or satisfying analytical quality requirements (sensitivity and specificity). During the pilot phase, NBS was implemented as a comprehensive programme encompassing much more than just laboratory testing. The concept involved close collaboration among private laboratories, Bavarian specialised centres of expertise (primarily university children’s hospitals) and the public health service in a framework of ‘Public–Private Partnership’ [12]. An official Newborn Screening Centre (NBS Centre) responsible for all of Bavaria was established within the public health service at the Bavarian Health and Food Safety Authority. The responsibilities of this NBS Centre include the coordination and quality management of the programme’s processes and development. The NBS Centre has repeatedly contributed to studies on the introduction of new target conditions and on other NBS issues. A central service hotline was established to support all those involved (parents, clinics, medical practices, midwives, specialised centres of expertise, and laboratories). The pilot project was accompanied by a population-based prospective long-term follow-up study which is still ongoing and aims to assess the long-term outcomes of children identified through NBS. Subsequent to the pilot phase, the NBS Centre has been maintained on a voluntary basis by the Bavarian State Government, thus ensuring its ongoing service to the Bavarian NBS programme.

Based on evidence from publications in the relevant scientific literature, the results of the Bavarian pilot project [14,15], and position statements from the Screening Committee of the German Society for Paediatrics and Adolescent Medicine [16], the national Federal Joint Committee (Gemeinsamer Bundesausschuss) decided to incorporate an expanded NBS programme into the national Paediatrics Directive (Kinder-Richtlinie) and thereby into the benefits catalogue of the statutory health insurance funds throughout Germany, effective since 1 April 2005 [17,18]. Subsequently, every newborn in Germany had to be offered NBS for an initial panel of 12 target conditions within the first 3 days of life. Meanwhile, the panel has been expanded to 17 target conditions [19]. The overall prevalence of the 17 target conditions is approximately 1 affected child per 730 screened newborns [20,21]. The prevalence of individual conditions varies, ranging from approximately 1 in 3000 to less than 1 in 700,000, thereby classifying these conditions as (very) rare [21,22].

Since 2005, candidate conditions have been screened in Bavaria alongside the established programme within a scientific study. The results of this study have been published [23,24] and will not be further discussed here. The present publication highlights the distinctive features of NBS in Bavaria, marked by a comprehensive programme and a centralised approach with systematic tracking and long-term follow-up. It summarises findings from 25 years of implementation and critically reflects on the programme’s strengths and limitations. Additionally, it reports quantitative and qualitative outcomes from the accompanying population-based follow-up study.

## 2. Materials and Methods

### 2.1. Framework of NBS in Germany

In Germany, target conditions, responsibilities, procedures, analytical methods, and quality assurance measures for NBS are defined in the Paediatrics Directive [19] and regulated by the Genetic Diagnostics Act [25]. This section and Figure 1 provide a synopsis of the German NBS process in accordance with the regulations effective during most of the observation period until 2024.

#### 2.1.1. NBS Process

Initially, parents must be informed about the NBS and provide written consent to the screening (informed consent). Following consent, a blood sample is taken from the newborn between 36 and 72 h of life, placed on a Dried Blood Spot (DBS) card, and sent to a laboratory accredited for NBS (NBS laboratory). If the newborn is discharged from hospital within the first 36 h of life or receives treatment that may affect the NBS results, blood should be collected for screening beforehand. In such cases, as well as for preterm infants born at or before 32 weeks of gestation, a repeat test (2nd DBS) is to be conducted after 36 h of life or after 32 weeks of gestation [19]. After analysis, the laboratory notifies the sender of the DBS card, typically the obstetrician, of the NBS results. Until 2024, positive screening results were to be communicated to parents by the sender of the 1st DBS. Depending on the suspected condition and the laboratory results, parents were then expected to either schedule an appointment for blood sampling or present the newborn at a specialised centre of expertise for confirmatory diagnostics [19,26].

#### 2.1.2. Laboratory Analysis

The specifics of laboratory diagnostics for NBS are regulated by the Paediatrics Directive [19] and generally align with internationally accepted analytical protocols [27]. Screening for hypothyroidism and CAH is performed using fluorescence immunoassay, biotinidase deficiency using fluorimetric activity determination, galactosaemia by means of enzymatic determination of total galactose after cleavage of galactose-1-phosphate and fluorimetric determination of galactose-1-phosphate uridyltransferase. Screening for other IEMs is performed using MS/MS and for severe combined immunodeficiencies (SCID) by measuring TRECs using quantitative or semi-quantitative polymerase chain reaction (PCR). Screening for sickle cell disease (SCD) is performed using high-performance liquid chromatography (HPLC) or PCR, and screening for 5q-associated spinal muscular atrophy (SMA) is performed using PCR to detect homozygous SMN 1 gene deletions. As demonstrated in the annual reports of the German Society for Neonatal Screening (DGNS) [28], the test kits and cut-off values vary between laboratories and have been repeatedly adjusted over the years.

A notable exception in international comparison is screening for cystic fibrosis (CF), which, unlike in most countries, is conducted in Germany using a three-tiered algorithm. This algorithm consists of a sequential series of tests for immunoreactive trypsin (IRT) and pancreatitis-associated protein (PAP), followed by a third molecular genetic stage that searches for 31 specific mutations of the cystic fibrosis transmembrane conductance regulator (CFTR) gene. The screening result is considered positive if the IRT value exceeds the 99.9th percentile, a process referred to as the “fail-safe” method or “safety net”, or if at least 1 of the 31 analysed mutations of the CFTR gene is detected in the molecular genetic analysis. In all other scenarios, the screening is considered negative [19].

### 2.2. Responsibilities of the Bavarian NBS Centre

In the German federal state of Bavaria, NBS is conducted in accordance with German NBS regulations [19,25]. Moreover, the central public health NBS Centre provides supplementary organisational structures and operational procedures. The following section and Figure 1 present a summary of the Bavarian-specific and additional NBS components, which are executed or coordinated through the NBS Centre.

#### 2.2.1. Programme Implementation, Coordination and Development

The Bavarian NBS Centre has been implemented to ensure the coordination and quality management of the programme’s processes and development. Since 2005, the responsibilities of the NBS Centre have encompassed the adaptation of the Bavarian structures to German NBS regulations. This involves committee work and communication with stakeholders, providing a service hotline, conducting or supporting studies, and developing and standardising informational materials and parental informed consent forms while ensuring adherence with data protection regulations.

Since 1999, parental consent within the Bavarian NBS programme has consisted of two parts: first, consent for the screening examination itself, and second, consent for the transmission of data to the NBS Centre. Since 2017, on the initiative of the NBS Centre and Bavarian stakeholders, parents have additionally been able to give consent for the direct transfer of data from the laboratory to a specialised centre of expertise in cases of positive screening results. Physicians at this centre then communicate the results and discuss subsequent steps with the parents.

#### 2.2.2. Ensuring Complete Coverage

With parental consent for data transmission, the screening laboratories transmit screening data via a secure connection to the NBS Centre. Since 2020, the NBS Centre also receives birth data from the Bavarian Municipal Data Processing Agency on a weekly basis [29]. These birth data are then automatically matched with the screening records. In accordance with the legally mandated health prevention counselling regulations in Bavaria [30], the NBS Centre contacts parents in cases where screening records are missing for their child.

#### 2.2.3. Tracking of Repeat Tests

In cases where follow-up information is not received within the predefined timeframe after a positive screening (one day for strong indications of disease reported by the NBS laboratory or two weeks for borderline NBS results), the NBS Centre proactively contacts the laboratory, the specialised centre of expertise, and/or the parents, either by phone or in writing, with repeated attempts if necessary (tracking). If these efforts prove unsuccessful, the Centre contacts local health authorities, who may visit the parents where appropriate. The Centre maintains follow-up until either subsequent tests provide negative results, or a diagnosis is established, ensuring continuity of medical care for the children [14]. Furthermore, the need for repeat tests—such as those required for blood samples taken before the completion of 36 h of life or 32 weeks of gestation, or due to medical interventions—is monitored as part of the tracking process. However, tracking is only possible with parental consent for data transmission.

#### 2.2.4. Ensuring Process Quality

NBS parameters such as birth prevalences of target conditions and processing times are routinely evaluated to ensure high process quality. This also allows for the identification of irregularities and areas for improvement to optimise NBS procedures. Evaluation results are shared with stakeholders and, where appropriate, disseminated via reports, newsletters, or scientific publications. In the event of repeated DBS quality issues or delays in DBS sampling or sending, etc., the NBS Centre contacts the responsible sender.

#### 2.2.5. Bavarian NBS Long-Term Follow-Up Study of Children Born 1999–2013

The NBS Centre is conducting a prospective population-based, long-term follow-up study across Bavaria, to which the families of children born 1999–2013 detected through NBS were invited, irrespective of where they receive medical care. The primary objective of this approach is to ensure that children identified through NBS receive appropriate treatment and competent medical care. The secondary objective is to evaluate the long-term effectiveness of NBS for each target condition, as well as its benefit in long-term outcomes.

### 2.3. Data Bases and Methods of Data Analysis

#### 2.3.1. NBS in Bavaria

The analyses included a comprehensive dataset spanning 25 years of NBS in Bavaria, comprising all children born and residing in Bavaria between 1 January 1999, and 31 December 2023. Personal and screening data were documented daily in an Access database until 2013, and since then have been recorded in an Oracle database. Data were subjected to thorough plausibility checks and regular analyses, similarly to the tracking data. Data on confirmatory diagnostics were obtained from the physicians or institutions providing treatment, contingent upon parental consent for data transmission. To establish the final diagnosis (whether to confirm or rule out the suspected diagnosis), data were evaluated for plausibility by the medical personnel at the NBS Centre and the screening laboratories, and, if necessary, validated according to predefined criteria in collaboration with clinical experts. The target parameters for quality assurance encompassed the coverage (screening participation rate), processing times, need for intervention by the NBS Centre, as well as birth prevalence, recall rate, and positive predictive value (PPV).

All contacts intended to enable the catch-up of missing screenings or to follow-up on positive findings were regarded as interventions by the NBS Centre. The analysed processing times included the interval from blood collection to the arrival of the DBS at the laboratory (i.e., delivery time), as well as the duration until the report was issued. In cases of positive screening results, the timeline further extended to cover the period until final diagnosis and treatment initiation. Newborns with false-negative and initially negative screening results—such as those observed in preterm infants born at or before 32 weeks of gestation—were excluded from the analyses of age at initial presentation at the specialised centre of expertise or treatment initiation. Target conditions for which the screening protocol schedules confirmatory diagnostics only at 2 to 3 weeks of age (SCD, CF) [26] were analysed separately in this context.

The recall rate—defined as the proportion of positive screening results (recalls) to the initial DBS samples—was calculated overall and stratified by target condition from all NBS data sets submitted to the NBS Centre. The positive predictive value (PPV), representing the probability of a condition being present given a positive test result, was calculated as the number of confirmed diagnoses divided by the number of recalls. Calculations were performed per target condition; thus, children with multiple suspected diagnoses may be counted multiple times. A separate PPV was also determined for findings classified by the laboratory as highly suspicious for the condition. False-negative screening results were included in the analyses if they were reported to the NBS Centre by February 2025. Confirmed diagnoses following a false-negative screening result or in cases where NBS was not performed were documented, as much as possible, using data from specialised centres of expertise and the Bavarian school entry examination. Coverage can be regarded as sufficiently high, albeit incomplete, to allow for the calculation of NBS specificity. In contrast, NBS sensitivity cannot be determined due to incomplete and non-systematic data capture, as well as the low prevalence of the target conditions.

The birth prevalence of each condition was calculated as the ratio of confirmed cases to the number of children screened for the respective condition in the NBS programme. Documented false-negative results and missed screenings were incorporated, where available. The prevalence of galactosemia was calculated using data solely from the classical form of the condition.

#### 2.3.2. Bavarian Prospective NBS Long-Term Follow-Up Study

The long-term follow-up cohort comprised children born and registered in Bavaria between 1999 and 2013 who were diagnosed with an NBS target condition, as defined in the Paediatrics Directive of 2004 [17]. The children are followed up until they reach the age of 18, whenever possible. The follow-up study was approved by the Ethics Committee of the Bavarian Chamber of Physicians (registration code 10099).

Only conditions requiring therapeutic intervention were considered (i.e., excluding mild hyperphenylalaninemia). Inclusion also required that, in the absence of NBS detection, treatment would have been initiated by 4 weeks of age. Exclusions encompassed syndromes or conditions potentially affecting development. In cases where hypothyroidism was found to be transient, follow-up was terminated and children with transient hypothyroidism were excluded from developmental analyses.

Upon parental consent, families were initially surveyed at the age of 1 year and subsequently at 2-year intervals. The NBS Centre employed detailed questionnaires to collect information on therapy and medical care, incidents of metabolic decompensation, the child’s physical, psychomotor, and emotional development, coping with the condition, basic sociodemographic data, life situation, and quality of life. From age 10, children were invited to provide consent for their own participation and were administered a separate questionnaire. The questionnaires primarily contained questions with predefined response options (quantitative analysis), alongside open-ended questions (qualitative assessment). In addition, medical records and clinical data were evaluated wherever available and used to validate or supplement data obtained from the questionnaires.

To examine the ethnic composition of the affected cohort relative to the overall Bavarian population, data from parental surveys regarding the mother’s country of origin were compared with the corresponding birth cohort from Bavarian perinatal surveys [31]. Experts in the respective condition evaluated the available written reports from the attending physicians in incidents reported by parents as metabolic crises. For this evaluation, metabolic decompensation was defined as any acute event resulting in overnight hospitalisation and involving biochemical derangement associated with the underlying condition, as confirmed by a medical expert.

Table 1 presents an overview of indicators and references used for analysing outcomes of the Bavarian NBS long-term follow-up study. In addition, adolescents were surveyed on condition burden and quality of life using the KIDSCREEN-10 score [32], while parents reported perceptions of burden over 18 years by means of a set of questions developed at the NBS Centre. To identify issues and challenges in communicating findings and managing the child’s condition, qualitative parental responses were analysed.

The parent-reported “Developmental Benchmarks”, which were used to assess psychosocial and motor development at age 6, are designed to be achieved by approximately 90% of children of the specified age [36,37]. The KiGGS survey, from which reference data for height [33], weight [34,35], behaviour [41,42], and quality of life [45] were drawn, is a comprehensive and representative German Health Survey for Children and Adolescents.

Due to the age structure of the cohort, occasional missing answers on the questionnaires, and some loss to follow-up over the years, the quantity of available data varies between outcome variables and generally decreases with increasing age.

Categorical data are presented as percentages with 95% confidence intervals, where appropriate. Deviations of standard deviation scores (SDS) from zero were assessed using one-sample Wilcoxon signed-rank tests (*p* < 0.05). To assess outcome changes over time in anthropometric measures up to age 10, developmental benchmarks at age 6, SDQ at age 8, and KiNDL-R at age 10, the long-term follow-up cohort was divided into three birth-year subgroups (1999–2003, 2004–2008, 2009–2013). Changes over time were then assessed using non-parametric Kruskal–Wallis tests for BMI and height SDS and KiNDL-R scale values, 95% confidence intervals for developmental benchmarks, and Chi-square or Fisher’s exact tests, as appropriate, for BMI and SDQ categories.

The data were managed in an Access^®^ database and analysed using SPSS^®^IBM (Version 29.0), R (Version 4.3.1), and RStudio (Version 2024.04.2-764).

## 3. Results

### 3.1. The Bavarian Newborn Screening Programme over 25 Years (1999–2023)

#### 3.1.1. Coverage

Between 1999 and 2023, a total of 2,895,385 newborn children were registered in Bavaria [46]. Of these, 2,854,190 newborns had NBS records in Bavaria, yielding a rate of 98.58%. At the onset of the NBS programme in Bavaria, this rate was 96.71%, and it subsequently remained consistently above 98%, exhibiting only minor fluctuations. In addition, the tracking of children without NBS records revealed that an estimated 1.25% of children had their NBS performed in a non-cooperating laboratory outside of Bavaria. Consequently, an NBS coverage rate of 99.83% can be deduced. To achieve this high coverage, parents of more than 4500 newborns were contacted annually due to missing data; and 100–200 children were subsequently rescreened each year. Approximately 75 test cards were lost annually in transit between the sender and the laboratory. Additionally, missed screening examinations were identified in cases of inter-facility transfers or early discharge from the hospital before 36 h of life. In all these cases, the NBS Centre re-initiated NBS as soon as possible and contacted the responsible senders of the DBS cards to identify and prevent errors on their part. Over the years, the number of parents refusing to share data with the NBS Centre has declined from approximately 2500 (2.45%) to approximately 300 per year (0.25%). A consistent number of around 140 parents (0.12%) each year has been reported to refuse NBS entirely.

#### 3.1.2. From Blood Collection to Result Reporting

The proportion of children screened within the first 72 h of life increased from 73.20% in 1999 to 98.12% by 2023. Similarly, the proportion of screening results communicated to the sender within 1 day of sample receipt in the laboratory increased from 44.50% in 1999 to 85.09% by 2023. In contrast, the proportion of DBS received in the laboratory within 1 day of blood collection decreased from 42.70% in 1999 to 25.81% by 2023. Meanwhile, the proportion of samples taking more than 3 days to reach the laboratory increased from 8.90% in 1999 to 23.21% by 2023. The NBS Centre communicates with DBS senders whose quality controls demonstrate long transit times and advises them on measures to avoid delays in their institution.

#### 3.1.3. Requested Repeat Screening Tests

During the study period, repeat tests were requested for 86,726 children. In 73,068 of these cases (84.25%), DBS samples were received by the laboratory without any further action. For the remaining 13,658 cases (15.75%), intervention by the NBS Centre was required, including 3754 requested recalls due to positive NBS results. Tracking was necessary for 21.68% of requested recalls due to positive NBS results in 1999 and for 11.68% in 2023. The local health authority was involved in 884 cases (1.02%). As a result of the tracking process, the number of completed repeat tests amounted to 85,935, corresponding to 99.09% of all requested repeats. A total of 791 parents (0.91%) could not be reached or declined testing, including only 54 cases following a recall. Without this tracking effort, 122 newborns with confirmed conditions would not have been diagnosed in time, despite a positive NBS result.

#### 3.1.4. Recall Rates

The recall rate decreased from 1.76% in 1999 for 12 target conditions to 0.39% in 2023 for 17 conditions (Figure 2). Between 2005 and 2007, an alternative test kit for CAH was employed, which may account for the increased recall rate at that time. The reduction observed in 2008 coincided with improved analytical methods and the implementation of a 2nd tier testing strategy for CAH. In 2017, the recall rate increased slightly again, coinciding with the introduction of additional NBS target conditions (Figure 2).

#### 3.1.5. Specificity and Positive Predictive Values (PPVs)

Screening specificity was consistently high across all target conditions, exceeding 99.9%. The overall PPV for all conditions included in the NBS programme between 1999 and 2023 was 12.36% (Table 2). In 2022/2023, the PPV for the 17 currently screened conditions was 32.67%, corresponding to approximately 400 positive results (recall rate = 0.40%) and 133 confirmed cases per 100,000 screenings. The PPV varied considerably between conditions (Table 2) and was markedly higher when the analysis was restricted to cases with where the NBS laboratory reported a strong indication of disease. In this subgroup, more than two thirds of the positive screening results were confirmed over the entire observation period, resulting in a PPV of 71.06% (Appendix A).

#### 3.1.6. Positive Cases, Birth Prevalence

Among 2,854,190 children with documented NBS, 2500 were diagnosed with a target condition, including 2 children with dual diagnoses, resulting in a total of 2502 positive findings. Overall, 1 in 768 newborns was affected by a target condition (Table 2), with a prevalence of 1 in 731 for the 17 conditions screened in 2022/2023. Of the affected children, 1354 were female (54.16%) and 1140 male (45.60%); gender information was unavailable in 0.24% of cases (n = 6). The observed sex distribution is primarily attributable to the higher prevalence of congenital hypothyroidism among females.

#### 3.1.7. Children Not Identified Through NBS

Over the course of 25 years, the NBS Centre was informed of 40 cases in which a target condition was not detected through NBS (false-negative, n = 35; NBS declined, n = 5; last column in Table 2).

In 3 of these cases, the diagnosis had already been confirmed prenatally due to a positive family history—specifically, 2 cases of CAH with maternal prenatal steroid therapy, and 1 case of GA I.

In 31 cases, a target condition was diagnosed despite an initially negative NBS result, either based on clinical symptoms or following a positive NBS result in a younger sibling. This group included 4 children with mild phenotypes (HPA, n = 1; biotinidase deficiency, n = 1; VLCADD, n = 2), and 10 children with known risk factors for false-negative results (CAH with mutation group B “I172N”, n = 8; intermittent MSUD, n = 1; GA I with low-excretor phenotype, n = 1). In 2 CAH cases, steroid treatment prior to sampling likely caused the false-negative screening outcome. All 6 children with false-negative hypothyroidism screening results were premature infants, including 2 monozygotic twins. In all these cases, either the repeat test at 32 weeks’ gestation or the first DBS at 33 weeks’ gestation yielded negative results. Among 7 children with false-negative CF screening results, 4 had an IRT value within the expected range, 2 had PAP values within the expected range (including 1 case with meconium ileus), and in 1 child, both disease-causing mutations were absent from the candidate gene panel while fail-safe was negative. NBS was false-negative with no identifiable explanation in 2 cases, 1 of galactosemia and 1 of PKU.

In 1 case of LCHADD, an initial positive NBS result was followed by false-negative confirmatory diagnostics.

All information on false negatives was transferred from the NBS Centre to the laboratories for review, and, if deemed appropriate, adjustment of cut-off values or algorithms.

Additionally, the NBS Centre was informed of 3 cases in which parents had declined NBS: 2 cases of hypothyroidism and 1 case of CF. Based on clinical symptoms, 2 further cases (hypothyroidism, n = 1; CF, n = 1) were diagnosed after the NBS had not been performed in accordance with the protocol in an out-of-hospital setting.

#### 3.1.8. Age at Initiation of Care or Therapy

A total of 2245 newborns were diagnosed with a target conditional that typically requires early intervention. Of these, data from 2102 newborns were eligible for the analysis of the age at which care, or therapy was initiated. Care was initiated within the first week of life for 1341 newborns (63.80%), within the first 10 days of life for 79.88% (n = 1679), and within the first 14 days for 87.35% (n = 11,836) (Table 3). When the laboratory reported a strong indication for the condition (n = 1552), care began within the first week in 68.69% of cases (n = 1322) and within the first 10 days in 85.18% (n = 1318). At the time of care initiation, 266 children were over 2 weeks old (Table 3). Of these, 67.67% (n = 180) exhibited borderline positive screening results or mild clinical phenotypes, resulting in one or multiple repeat tests prior to referral to confirmatory diagnostics. Early therapy within the first week of life was achieved for 56.0% of children during the pilot phase of the Bavarian NBS programme (1999–2003), and for 69.0% in the most recent years studied (2019–2023), while the proportion of children receiving care after more than 2 weeks of age was 18.4% in the first and 11.4% in the most recent years.

In certain cases, presentation at the specialised centre of expertise was delayed. For example, 1 MCADD patient’s DBS sample was lost en route to the laboratory, and, in another case, the involvement of the child and youth welfare agency was required. In both instances, timely initiation of care by 15 days of life was successfully achieved due to the active support provided by the NBS Centre.

During the study period, 246 children were diagnosed with conditions not requiring urgent diagnostic or therapeutic intervention (CF, n = 205; SCD, n = 41). For 147 children diagnosed with CF (71.70%), treatment was initiated within the targeted first 4 weeks of life (Appendix A), and for 36 children diagnosed with SCD (87.80%) within the targeted first 3 months of life.

#### 3.1.9. Setting of Confirmatory Diagnostics

Confirmatory diagnostics were performed at specialised centres of expertise in 1805 affected children (72.20%). Newborns diagnosed with endocrinological conditions were less frequently referred to such centres (hypothyroidism, n = 408, 45.38%; CAH, n = 127, 58.53%), whereas 259 MCADD cases (87.50%) were confirmed in specialised centres (Appendix A).

### 3.2. Results of the Bavarian NBS Long-Term Follow-Up Study (Birth Cohort 1999–2013)

#### 3.2.1. Study Population

A total of 1240 children born and registered in Bavaria were diagnosed with an NBS target condition in the birth cohort from 1999 to 2013, on which the long-term study is based (Appendix A). Tracking was continued for all these children until it was ensured that they were receiving appropriate ongoing therapy and medical care.

The maternal country of origin was documented for 1153 children. Of these, 807 mothers (70.0%) were of German origin—approximately 10% less than the average in the Bavarian general population between 1999 and 2013 (80.7%). Mothers from the Middle East or North Africa accounted for 12.3% (n = 142), markedly more than the 4.3% observed in the reference population. Parental consanguinity was reported for 61 children of the long-term follow-up study.

Among the 1240 children diagnosed with a target condition 990 children met the inclusion criteria for the long-term study and their families were contacted from the NBS Centre. A total of 903 families participated in the study (91.2%). Data from 51 children in whom permanent hypothyroidism was ruled out during the study period were excluded from further analysis. The overall loss-to-follow-up rate over 25 years was 10.2% (n = 92). By the end of 2024, final or current follow-up information was available for 811 children, representing 81.9% of the study cohort. Stratified analyses revealed no evidence of changes in outcome findings over time during the course of the observation period. Further details on follow-up participation are provided in Figure 3, Table 4 and Appendix A

#### 3.2.2. Medical Care

Except for hypothyroidism, which was managed solely in paediatric practices for 36.6% of affected children (124/339), 94.5% of children with other diagnoses (484/512) were seen at least once at a specialised centre of expertise. By age 10, 80.9% (360/445) of these children continued to receive medical care at these centres. In cases where the survey revealed potential gaps in care—such as reported absence of recent visits to specialised centres of expertise or missing emergency cards—parents were informed by the NBS Centre about current care recommendations and the nearest accessible specialised centres of expertise.

#### 3.2.3. Decompensation Episodes

Of the children participating in the follow-up study, 374 were diagnosed with conditions that predispose to metabolic or electrolyte decompensation. No decompensation episodes were reported in 279 of these cases (74.6%, Table 5). Episodes of metabolic or electrolyte decompensation were documented in 95 children with IEMs or CAH. Of these children, 84 experienced a single decompensation episode, while 11 had multiple episodes.

In 45 children, decompensation occurred during the neonatal period. This group includes 8 newborns with a neonatal metabolic decompensation before the NBS result was available (3 with MCADD, 2 with VLCADD, 1 with MSUD, 1 with CACT, and 1 with CPT I) and 9 newborns with classic galactosemia who had already shown coagulation disturbances at diagnosis, with varying additional clinical symptoms and biochemical deviations. Furthermore, 28 children diagnosed with CAH suffered from an electrolyte imbalance with clinical symptoms (electrolyte crisis).

Post-neonatal decompensation episodes occurred in 40 of the affected children during infections (gastrointestinal infections, n = 27; other febrile infections, n = 13), and predominantly until the age of 4, with only 7 episodes occurring at ages 5 to 17 (diagnoses: CAH, MSUD, MCADD, VLDADD, LCHAD, IVA). In some cases—particularly in CAH (n = 21) and MCADD (n = 6)—there is evidence that established emergency protocols were not fully implemented or only partially followed prior to the decompensation episode, either by family members or healthcare personnel. Decompensations occurred irrespective of whether children were under care at specialised centres of expertise. Notably, only 2 children with MCADD and 6 with CAH who experienced decompensation episodes were not managed in specialised centres of expertise.

During decompensation, 7 children died (for diagnoses and details, see Table 5 and Appendix A), including 5 within the first year of life, 1 at 22 months (CPT I), and 1 at age 3 (MCADD). 1 child with MSUD, who had previously suffered multiple decompensation episodes, died at age 13 following a liver transplant. Nevertheless, the majority of decompensation episodes were managed promptly and effectively.

#### 3.2.4. Height and Body Mass Index

Overall, growth and body mass index (BMI) values in the study population were comparable to those of age-matched children in Germany (Figure 4 and Figure 5. Children with CAH showed altered growth trajectories, often characterised by reduced height (previously published [47]) (Appendix A). A similar trend toward shorter stature was observed in children on protein-restricted diets (Appendix A).

At age 8, underweight was present in 10% of the study population (CI 95, 7.9–12.5%) compared to 7.8% in the reference population (CI 95, 6.9–8.8%), while obesity was observed in 7.6% (CI 95, 5.7–9.8%) versus 6.4% (CI 95, 5.5–7.3%), indicating a slightly but not significantly higher prevalence of both. Underweight particularly affected IEMs, while obesity was especially notable in CAH ([47], Figure 5 and Appendix A).

#### 3.2.5. Developmental Benchmarks and School Careers

As shown in Table 6 and Appendix A, and Appendix A, the evaluation of developmental benchmarks up to the age of 6 and school careers up to the age of 14 revealed no deviations from the Bavarian reference population for the most common NBS diagnoses: hypothyroidism, CAH, PKU, and MCADD. However, children with other, rarer IEMs showed developmental or motor problems more frequently (Table 6 and Appendix A) and attended special needs schools more often (Appendix A). This primarily affected children with galactosemia, MSUD, LCHADD, and carnitine cycle defects, as well as a few children with VLCADD and GA I, but not those with biotinidase deficiency or IVA (Appendix A).

#### 3.2.6. Behaviour and Quality of Life in Children and Adolescents

The proportion of children exhibiting behavioural difficulties as assessed by the SDQ at age 8 did not differ from that of the German reference population (KiGGS survey, Appendix A) with respect to the total score or the subscale scores for emotional symptoms, conduct problems, hyperactivity, peer relationship problems, and prosocial behaviour. Likewise, health-related quality of life assessment using the KINDL-R instrument at age 10 revealed a high degree of similarity between the study population and the German reference population from the KiGGS survey (mean KINDL-R scale value [CI 95%]: Bavarian NBS follow-up study: 79.9 [79.1–80.7], KiGGS survey: 79.0 [78.7–79.3]).

Despite reporting condition-specific challenges during puberty—such as dietary restrictions and medication regimens—adolescents consistently rated their overall quality of life and health as good, a perception that persisted through to the final follow-up at age 18.

#### 3.2.7. Summary of Long-Term Follow-Up Outcomes

With certain notable exceptions, children diagnosed and treated early through NBS showed outcomes analogous to the reference population. Table 7 presents an overview of main findings from the ongoing long-term follow-up study on the development of children born between 1999 and 2013 in Bavaria and diagnosed with an NBS target condition.

#### 3.2.8. Challenges and Burdens Experienced by Parents

In their questionnaires, around 25% of the parents reported experiencing anxiety and uncertainty following the notification of a positive screening result. This distress was largely attributed to the way information was conveyed and the lack of clear, comprehensive explanations at that time. Parents expressed fears and initial difficulties in managing their child’s condition and adhering to therapy regimens. Over time, however, these concerns diminished after medical care and therapeutic support had been established. In subsequent surveys, 95% of parents repeatedly reported that they were coping well with their child’s condition. Nevertheless, 42% of respondents mentioned specific concerns in the questionnaires, including the following:Considerably long distances to specialised centres in a large federal state like Bavaria.Occasions where medical personnel outside specialised centres of expertise appeared insufficiently informed about NBS conditions and did not utilise emergency protocols provided by the patients or their family. This was perceived as alarming, especially in emergency situations that may not be directly related to the condition, such as accidents, yet require interventions specific to the condition.Difficulties implementing therapeutic recommendations in daily life, such as ensuring adherence to diet and medication during full-day care or school excursions. Additionally, consistently carrying emergency cards posed a challenge for some children.Financial burdens associated with certain conditions, including the cost of specialised foods not reimbursed by health insurance.

The analysis of responses from the parents of 289 participants who had reached the age of 18 until 2024 (birth cohorts 1999–2006) indicated that the perceived burden associated with their child’s endocrine or metabolic condition generally decreased over time. However, in some cases, concerns about the future appeared to lead to a renewed increase in perceived burden (Figure 6).

## 4. Discussion

Newborn screening (NBS) using dried blood spots (DBS) is conducted worldwide, but in various countries, including Germany, it has not been part of an integrated public health programme with centralised coordination, as recommended by the WHO [48], EU experts [49], and numerous publications [3,50,51,52,53,54]. In Bavaria, NBS has been implemented as a programme for the past 25 years, with a central NBS Centre (Public Health Service) responsible for coordination, quality assurance, and evaluation. The work of the NBS Centre, together with the expertise of the NBS laboratories in analytics, ensures the Bavarian NBS programme’s high quality. The NBS Centre’s consistent tracking activities achieved almost complete coverage and resolve rates. Further, the coordination and committee work of the NBS Centre with disease experts, laboratories, and other stakeholders enabled the straightforward and uncomplicated introduction of new target diseases and the implementation of a centre-independent, population-based long-term study.

### 4.1. Experiences from 25 Years of the Bavarian Newborn Screening Programme

#### 4.1.1. Coverage

A high NBS coverage rate consistently exceeding 99.8% was achieved in Bavaria, following an introductory development phase that saw a slightly lower rate in the early years. Given a prevalence of approximately 1 in 730 for the 17 currently screened conditions and an annual birth cohort of around 110,000, about 150 newborns are diagnosed with an NBS condition each year. This implies that, statistically, in Bavaria alone, one case per year would be missed if NBS coverage dropped below 99.3%. Beyond the avoidable suffering for the affected individuals and their families, undiagnosed cases also carry substantial health economic implications, given the potential for lifelong disabilities. Therefore, ensuring comprehensive participation in this public health measure remains essential. Nevertheless, the administrative effort required—over 4500 parental notifications for 100–200 catch-up screenings each year—raises the question of whether this method of ensuring complete coverage is still appropriate. Without a name-based or digital matching system, it is not possible to ensure that every child is screened or to calculate a valid coverage rate. Digitalisation of the NBS process might enable the most effective follow-up for the children initially missed.

The Bavarian data indicate that, at present, over 99% of parents provide informed consent for both NBS and data transmission to the NBS Centre, demonstrating broad acceptance not only of NBS itself but also of a comprehensive programme with a centralised structure for systematic tracking.

#### 4.1.2. Tracking of Requested Repeat Tests

Another strength of the Bavarian programme is the active follow-up of requested repeat tests by the NBS Centre. Through its intervention, approximately 99% of repeat tests were resolved, and over a 25-year period, only 54 out of 19,914 recalls remained unresolved. Tracking was particularly important for 122 affected children to ensure timely completion of confirmatory diagnostics. A comparison of these findings with previous data from across Germany, which indicate that approximately 20% of requested repeat tests are not followed up [22], underlines the importance of effective tracking, as similarly reported in the USA and Sweden [6,55,56]. As a consequence of these findings, the Paediatrics Directive was amended from January 2025 [19] to include regulations on the tracking of requested repeat tests nationwide. Accordingly, both screening laboratories and centres must now verify the follow-up of repeat tests and appointment adherence, and contact parents if delays occur.

Digital transformation—including the implementation of a unique nationwide screening ID and a centralised digital platform hosted on a secure server—could substantially reduce the administrative burden and simplify processes. This platform would be accessible to screening laboratories and centres, specialised centres, paediatricians, and parents, featuring role-based access control and a data protection-compliant key management system [57,58]. Automated, case-specific reminders could be sent to the personnel responsible for tracking or, when appropriate, directly to parents, thereby enhancing the completeness and timeliness of screening and follow-up.

#### 4.1.3. Processing Times

Throughout Germany, as demonstrated by the analysis of national German NBS data [22], treatment is initiated within the first two weeks of life for 79% of children, while in the Bavarian NBS programme, systematic tracking enabled the initiation of treatment within the first 14 days of life in approximately 87% of infants with time-sensitive conditions. Nonetheless, there are accounts of 17 cases in which metabolic decompensation occurred before positive screening results could be communicated and thus could not be prevented. These occurrences, as well as electrolyte imbalances in CAH, are well documented in the literature [59,60,61] and appear unavoidable in some conditions, even in well-established NBS programmes with early blood sampling between 36 and 72 h and systematic tracking.

The issue of increased delivery times for DBS cards is a matter of concern both in Bavaria and throughout the whole of Germany [22,28]. Most DBS cards are transported via postal services, a method that remains straightforward and economical, but the postal conditions and transport times have changed considerably over the years. However, the effective quality management of the NBS Centre, coupled with recommendations to take DBS samples as early as possible within the stipulated time frame, has contributed to the maintenance of relatively constant ages at the initiation of care. Consequently, the current delivery times have been tolerated to date. In the event of future deterioration, alternative arrangements for the transportation of DBS cards may need to be considered, along with the securing of their necessary financing.

#### 4.1.4. Quality Criteria of the Bavarian NBS Programme

Thanks to high analytical quality, in approximately two-thirds of cases with positive screening results in 2023, the suspected diagnosis could be definitively ruled out following repeat testing or confirmatory diagnostics, although reassurance rates vary by condition and biomarker levels. In cases where the NBS laboratory reported strong indications for an NBS target condition, the PPV exceeded 71.43%, indicating that two-thirds of these cases were confirmed as true positives. This enabled the reduction in both unnecessary anxiety among families and the number of confirmatory tests required. These findings compare favourably with other public health screening programmes, such as newborn hearing screening, where PPVs of 1.5% [62] and 4.2% [63] have been reported.

Despite this high level of analytical quality, 35 children with false-negative NBS results were reported to the NBS Centre. As NBS generally aims to ensure that no affected newborn is overlooked, information on false-negative NBS results is of particular importance for the continuous improvement of NBS procedures, cut-off values and algorithms. Conversely, it is imperative to acknowledge that, even with meticulous efforts, false negative results can still occur with screening processes. In clinical practice, target conditions of NBS must always be considered in the differential diagnosis when typical symptoms are present—even if the NBS was negative. In instances where no explanation for a false-negative NBS result can be identified, it cannot be excluded that another person’s blood might have been tested [64]. Known causes of false-negative results include the 172N mutation [65,66], which is associated with delayed elevation of 17-hydroxyprogesterone (17-OHP), and steroid therapy administered prenatally or postnatally in cases of CAH. Furthermore, negative results of repeat tests after an initial positive screening result have been documented in cases involving anabolic metabolic states, such as lipid metabolism disorders, as well as in mild or intermittent variants of IEMs [67,68,69,70]. False-negative screening results in CF, due to the absence of elevated immunoreactive trypsinogen (IRT) or pancreatic-associated protein (PAP), or very rare mutations, are also regularly reported [71]. Among children with false-negative hypothyroidism screening results, 2 pairs of monozygotic twins were identified. The phenomenon of masked hypothyroidism in NBS caused by fetofetal transfusion in monochorionic twins—where the euthyroid co-twin conceals the diagnosis—has been well documented [72,73,74], suggesting that an adjustment of the NBS algorithm for twins should be considered.

Since children with NBS target conditions are not systematically or comprehensively documented in a registry in Bavaria or across Germany, it remains uncertain whether additional affected children have been missed by the NBS programme. Consequently, there is an urgent need to establish a comprehensive registry mandating the reporting of cases undetected by NBS. Examples include the systematic documentation of later-detected false-negative cases, as implemented in the Netherlands through the confirmatory diagnostics database [75] and in Sweden through the registry for congenital metabolic disorders [76]. At present, the sensitivity of the NBS programme in Bavaria can only be estimated. Beyond enabling sensitivity calculations, integrating an optimally designed registry into the digitalization of the NBS process has the potential to substantially enhance overall programme quality.

#### 4.1.5. Confirmatory Diagnostics and Notification of Positive NBS Results

University hospitals in Bavaria were involved early on in the conceptualisation of the Bavaraian NBS programme. This allowed for the designation of specialised centres of expertise for all target conditions based on defined criteria. These centres work closely with the NBS Centre and the cooperating NBS laboratories and have performed over 70% of confirmatory diagnostics at an early stage. This structure and the close collaboration are crucial elements for the success of the NBS programme, ensuring timely and guideline-adherent confirmatory diagnostics of positive screening results, as well as early initiation of therapy when necessary and expert, communicative support for families from the outset. This is especially important for the very rare target conditions of NBS [77,78,79]. The establishment of such centres aligns with recommendations from the European Council’s policy on rare diseases [79,80].

Positive laboratory results in the context of NBS do not, in principle, allow for a definitive diagnosis; rather, they may serve as an indication for a condition that must always be further investigated in a recall process [26,27]. It is considered optimal for parents to be first informed of a positive NBS result by a specialised centre, given parental consent. As demonstrated in various studies [81,82], this approach, which has been a well-received practice in Bavaria for years, facilitates earlier diagnosis and reduces parental anxiety. In the 2024 revision of the German Paediatrics Directive [19], improvements were made to the NBS notification procedures when compared to previous iterations. However, despite the evidence and explicit recommendations of the German Genetic Diagnostics Commission [83], the initial notification of positive NBS findings to parents directly through specialised centres was not incorporated into the current Paediatrics Directive and thus has not yet been implemented on a nationwide basis across Germany.

#### 4.1.6. Birth Prevalences

Based on more than 2.8 million screened newborns, birth prevalence estimates are robust and consistent with national and European data [21,22,84,85,86,87,88,89,90]. However, prevalences are influenced by the ethnic composition of the population and the proportion of consanguineous families, which is notably high in regions such as the Middle East, North Africa, and Sub-Saharan Africa [91]. In these regions, the prevalence of genetically determined conditions, including IEMs, is particularly high [92]. This is reflected in the higher proportion of migrants among children diagnosed with target conditions compared to the overall Bavarian population.

### 4.2. Bavarian NBS Long-Term Follow-Up Study

Routine documentation and evaluation—including long-term follow-up and health economic assessments [49,93]—are essential for the continuous evaluation and quality improvement of NBS programmes [9,57,94]. However, to date, we are not aware of any population-based study that has examined long-term outcomes over an 18-year period, providing a comprehensive overview of the metabolic and endocrinological target conditions identified through NBS. Aside from publications focusing on a single condition, only a few studies have analysed the long-term course in children with metabolic target conditions managed in specialised centres of expertise; however, none have done so on a population-based scale [10,61,95].

The Bavarian long-term follow-up study of children born between 1999 and 2013 employed a prospective, population-based observational design. This included regular parental surveys as well as analysis of medical reports and questionnaires, aiming to include families of all children identified through NBS, regardless of whether they received treatment at a specialised centre of expertise or elsewhere. The initial plan was to obtain the relevant data from the treating physicians; however, this could not be conducted due to low participation rates among physicians.

The initial participation rate exceeded 90%, approximately 20% higher than that reported in a comparable study on IEMs conducted in Southwest Germany [10]. A key strength of the Bavarian study is its population-based design with low-threshold participation via parental questionnaires, in contrast to other long-term follow-up studies of children with IEMs, which typically require clinical examinations and the involvement of multiple centres [10,61].

#### 4.2.1. Results of the Bavarian NBS Long-Term Follow-Up Study

Despite early diagnosis and treatment, 23% of children with a condition that frequently leads to decompensation if untreated experienced one or more episodes of decompensation, with 7 of these cases resulting in death, even though they were cared for in specialised centres of expertise.

The proportion of Bavarian patients known to have experienced metabolic decompensation was 24.7% for all conditions at risk (including CAH), and 16.4% for IEMs at risk only. These figures are clearly lower than those reported in the literature for unscreened cohorts. For example, decompensation rates ranging from 62.5% to 95.0% have been reported for MCADD patients who have not undergone NBS [96,97,98,99]. This corroborates the anticipated benefits of early detection through NBS. In comparison with the limited data on previously reported proportions of screened IEM patients having experienced decompensations, namely, 13.0% [100] and 25.7% [10], the rate observed among the Bavarian patients can be considered as relatively low. These differences may be due to variations in the definitions employed or random fluctuations related to the rarity of both conditions and decompensations. However, the possibility of a methodologically induced partial underreporting of metabolic decompensations in the Bavarian cohort cannot be entirely excluded.

The mortality rate of the Bavarian cohort for children at risk of decompensation is 3%, which lies within the range of rates reported for comparable screened cohorts (1.1% to 5.9%) [10,101,102,103]. Studies of unscreened cohorts have found mortality rates of 12.5% to 25% for MCAD deficiency [96,97,98,99] and 22.4% for a larger unscreened IEM cohort [61], which again impressively confirms the benefits of NBS.

At age 6, psychomotor development—assessed through parental reports of developmental benchmarks—was generally comparable to that of the Bavarian reference population [37]. The findings for children diagnosed with IEMs align with those obtained using the Denver Developmental Screening Test (DDST) in the previously mentioned study [10]. In both studies, poorer developmental outcomes were more frequently observed in children with the same diagnoses. Similarly, regarding height and body mass index (BMI), the findings are consistent across both studies and indicate a tendency toward shorter stature in children with conditions requiring protein-restrictive dietary management [104].

For children diagnosed with hypothyroidism, the findings confirm expectations of normal physical and cognitive development when treatment is initiated early and maintained consistently [105,106,107]. In children with CAH, the prevention of severe neonatal crises and fatalities could be achieved and the children diagnosed with CAH demonstrated good physical and psychomotor development, along with a positive quality of life. However, they tended to be shorter than their healthy peers and showed a higher prevalence of overweight or obesity. These findings, which have been published separately [47], are considered in current treatment recommendations and remain an area of ongoing research [108,109].

Encouragingly, both parental assessments of their children’s behaviour and health-related quality of life, as well as the children’s own responses, indicated positive outcomes. The burden experienced by parents due to the condition decreased over the years. Similar findings have been reported in the study from Southwest Germany on children diagnosed with IEMs [110].

Despite the overall favourable findings regarding development and quality of life, the child’s condition remained a significant challenge for families and affected their daily lives, similar to other rare conditions [79,80]. At age ten, approximately 80% of children—excluding those with hypothyroidism—remained under the care of specialised centres of expertise. However, distances to such centres are often considerable, and for many families, the transition to adult care remained unclear. Prior to the implementation of NBS, children with these rare conditions often did not survive into adulthood or remained undiagnosed. Consequently, knowledge about these conditions remains limited in adult medicine. In this context, the recommendations of the German Ethics Council for rare diseases also apply to NBS target conditions. These include the establishment and appropriate funding of specialised centres for adults, the implementation of patient registries to generate evidence on long-term outcomes and identify NBS failures, the promotion of targeted research, and the ongoing training of healthcare professionals, with a focus on raising awareness of the specific needs of individuals with rare conditions [79].

#### 4.2.2. Overall Evaluation of the Long-Term Follow-Up Study, Strengths and Limitations

In summary, the results of the long-term follow-up study demonstrate that nearly all children detected through NBS in Bavaria receive ongoing medical treatment and care, and that NBS has contributed to normal development in most affected children. Limitations apply to individual cases and specific diagnoses. Nonetheless, the findings also highlight persistent challenges that require further attention. In certain cases, the NBS Centre, in collaboration with specialised centres of expertise and screening laboratories, has been able to respond accordingly. For children from families with additional support needs, assistance from child and youth welfare services can be crucial for successful therapeutic outcomes. An NBS Centre facilitates this process through continuous communication with families, contingent on their consent.

A distinctive strength of the Bavarian study lies in its population-based design, encompassing an 18-year observation period and a large proportion of data collected directly from affected families rather than solely from clinical records. While this approach may pose potential limitations regarding data validity, it also contributes to a high participation rate, initially exceeding 90% and remaining around 80% over the long term. Comparison with results from the multicentre Southwest German study, which included clinical examinations, revealed highly concordant findings [10,104,110]. These results challenge the commonly held assumption that parent- or patient-reported outcomes are inherently unreliable or biassed, instead suggesting that standardised and validated parent or patient questionnaires may provide a sufficiently accurate representation of reality. The combined use of questionnaires and medical records thus represents a low-threshold, cost-effective method for programme evaluation. Despite the high participation rate, selection bias can be assumed, e.g., due to language difficulties or unfavourable family situations.

Nevertheless, clinical examinations remain indispensable for in-depth clinical topics, individual assessment, and risk stratification of disease progression and outcomes. Further publications presenting refined analyses of the Bavarian long-term data on specific target conditions are forthcoming. Given the rarity of all NBS target conditions, the conclusions drawn—despite the population-based approach and high participation rates—are limited by small sample sizes within subgroups. This underscores the need for interregional or international collaborative follow-up studies and patient registries [2,48,111].

## 5. Conclusions

The results of this 25-year evaluation of NBS in Bavaria demonstrate that the concept of NBS as a comprehensive programme integrating a centralised NBS Centre for coordination, quality assurance, and evaluation in close collaboration with screening laboratories and specialised centres facilitates near-universal participation and comprehensive follow-up of positive NBS results. Thanks to the intervention of the NBS Centre, high rates of NBS coverage, positive NBS clarification and early treatment initiation could be achieved for Bavarian children. In instances such as the loss of test cards, the omission of NBS or requested repeat screening tests, or delayed dispatch, the effort involved in identifying the cause and communicating directly with the responsible party largely proved worthwhile. Should consistent, specific quality measures prove ineffective, systemic changes may be proposed. The improvements achieved through the Bavarian NBS programme after the initial years, as well as comparisons with NBS data from across Germany, demonstrate the advantages of quality management and tracking strategies. When implemented and coordinated based on the existing structures and ideally supported by digital instruments, these strategies have the potential to further improve NBS in other regions, for the best interests of affected children.

The results of the population-based long-term follow-up study demonstrate that nearly all children detected through NBS in Bavaria receive ongoing medical treatment and care and that NBS has contributed to normal development in most affected children. These findings underscore the significant benefits of NBS for affected children and their families, while also highlighting existing challenges and the need for continued research.

## Figures and Tables

**Figure 1 IJNS-11-00114-f001:**
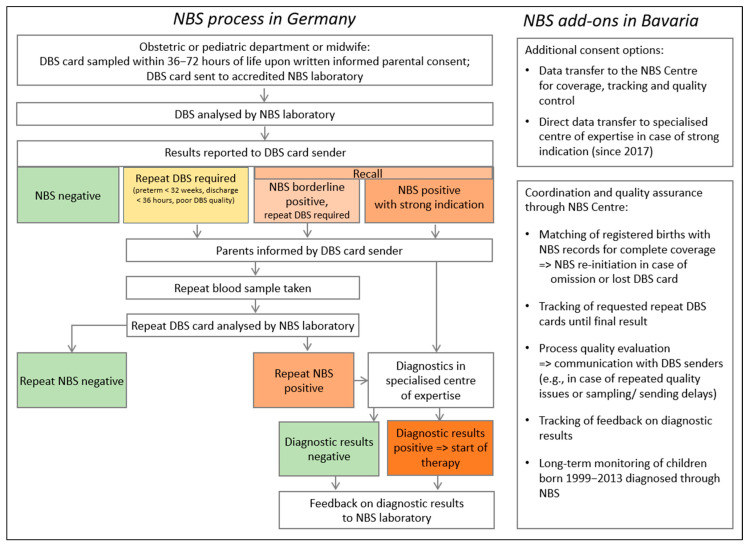
Overview of the NBS process in Germany until 2024 and the Bavarian NBS programme’s additional components. Abbreviations: DBS, dried blood spot; NBS, newborn screening.

**Figure 2 IJNS-11-00114-f002:**
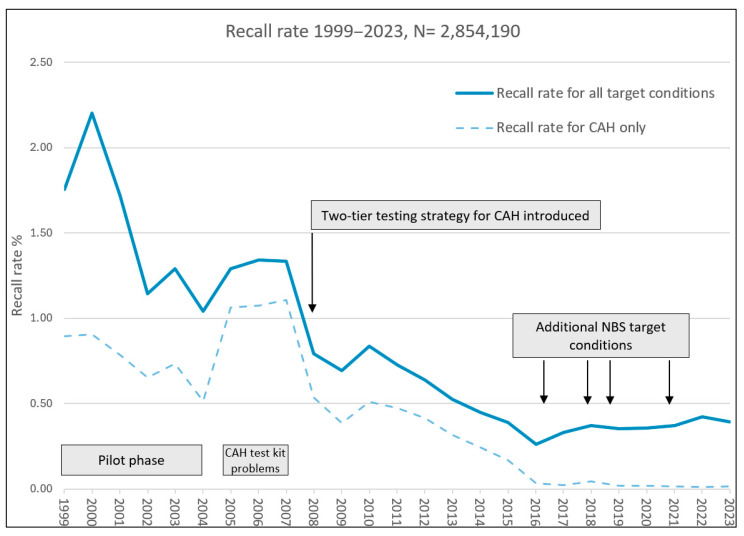
Recall rates from 25 years of the Bavarian NBS programme (1999–2023; n = 2,854,190). Abbreviations: CAH, congenital adrenal hyperplasia; NBS, newborn screening.

**Figure 3 IJNS-11-00114-f003:**
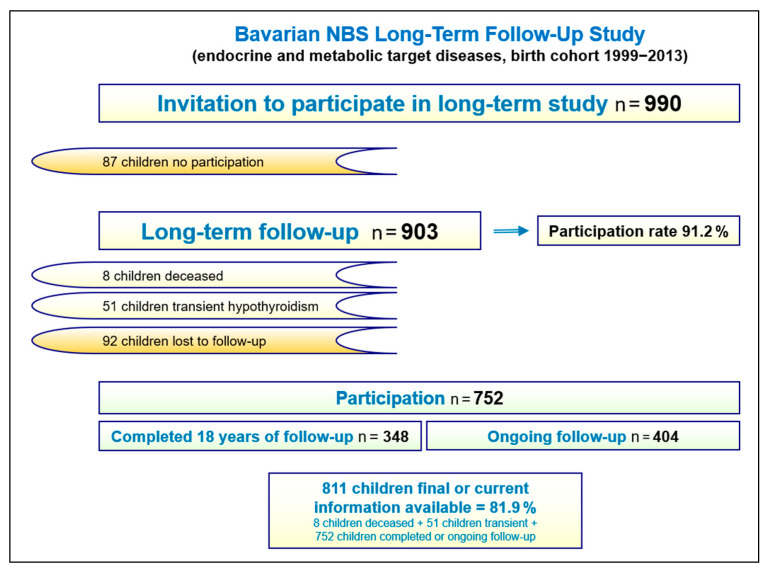
Participation in the Bavarian NBS long-term follow-up study (birth cohort 1999–2013). Further details are provided in Appendix A.

**Figure 4 IJNS-11-00114-f004:**
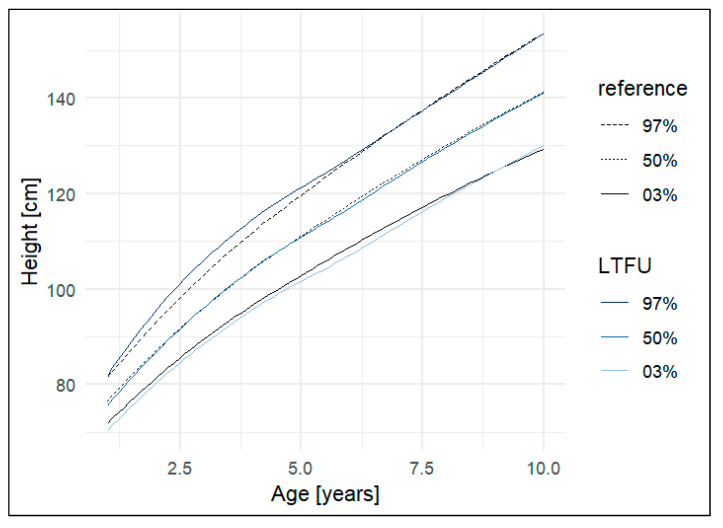
Height development until age 10 in participants of the Bavarian NBS long-term follow-up study (LTFU, birth cohort 1999–2013) compared with the German health survey for children and adolescents (KiGGS) (Reference) [33]. Abbreviations: LTFU, long-term follow-up study.

**Figure 5 IJNS-11-00114-f005:**
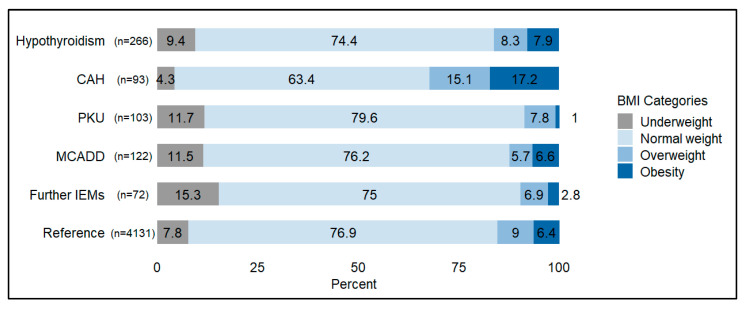
BMI categories at age 8 in participants of the Bavarian NBS long-term follow-up study (birth cohort 1999–2013, BMI data available at age 8 for 656 participants) and the German health survey for children and adolescents (KiGGS) (Reference) [34]. Abbreviations: BMI, body mass index; CAH, congenital adrenal hyperplasia; IEMs, inborn errors of metabolism; MCADD, medium-chain acyl-CoA dehydrogenase deficiency; PKU, phenylketonuria.

**Figure 6 IJNS-11-00114-f006:**
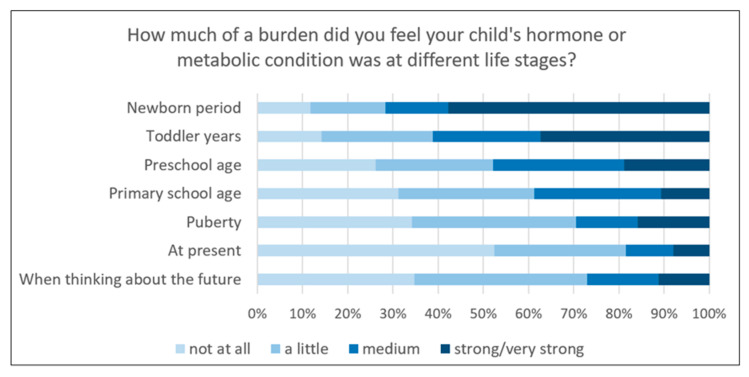
Responses given 2017–2024 from parents of 18-year-old participants regarding the perceived burden caused by their child’s condition over time (birth cohort 1999–2006; n = 289).

**Table 1 IJNS-11-00114-t001:** Overview of indicators and references for main outcome categories of the Bavarian NBS long-term follow-up study.

Outcome Category	Outcome Indicator	Reference Population
Growth	Height development until age 10 Height SDS until age 10	KiGGS survey [33]
BMI	BMI categories at age 8 BMI SDS until age 10	KiGGS survey [34,35]
Development	Preschool development: Developmental benchmarks [36] at age 6	Bavarian school entry examination [37]
	School career until age 14	Bavarian State statistics [38]
Behaviour	Strengths and Difficulties Questionnaire (SDQ) [39,40] at age 8	KiGGS survey [41,42]
Quality of Life	Health-Related Quality of Life (HRQoL) at age 10 (KINDL-R instrument) [43,44]	KiGGS survey [45]

Abbreviations: BMI, body mass index; KiGGS, German Health Survey for Children and Adolescents; SDS, standard deviation score.

**Table 2 IJNS-11-00114-t002:** Screening outcomes and metrics by diagnosis in the Bavarian NBS programme (1999–2023; N = 2,854,190).

Condition	Cases	Prevalence	Recalls	Recall Rate	PPV	False-Negative/NBS Declined
	n		n	%	%	n
Hypothyroidism	899 ^6^	1:3175	2100	0.07	42.33	6/3
CAH	217 ^7^	1:13,153	11,881	0.42	1.73	12
Biotinidase deficiency	54	1:52,855	599	0.02	8.85	1
Classic galactosemia	44	1:64,868	1410	0.05	3.05	1
PKU/mild HPA	577 ^6^	1:4947	1076	0.04	53.44	2
MSUD	12	1:237,849	62	<0.01	17.74	1
MCADD	297 ^7^	1:9610	441	0.02	67.35	
LCHADD	15	1:190,279	116	<0.01	12.93	1
VLCADD	31	1:92,071	191	0.01	15.18	2
Carnitine cycle defects ^1^	8	1:356,774	98	<0.01	8.16	
GA I	24	1:118,925	176	0.01	12.50	2
IVA	25	1:114,168	491	0.02	5.09	
Tyrosinemia type I ^2^	3	1:242,151	33	0.01	9.09	
SCID ^3^	8	1:68,875	157	0.03	5.10	
SCD ^4^	41	1:6516	77	0.03	53.25	
SMA ^4^	33	1:8096	35	0.01	94.29	
CF ^5^	214	1:4256	971	0.11	21.11	7/2
Total	2502	1:768	19,914	0.84	12.36	35/5

^1^ Carnitine cycle defects: CPT I, n = 3, prevalence 1:951.397; CPT II, n = 3, prevalence 1:951.397; CACT, n = 2, prevalence 1:1.427.095. ^2^ NBS target condition since March 2018 (N = 726,453). ^3^ NBS target condition since September 2019 (N = 550,998). ^4^ NBS target condition since August 2021 (N = 267,171). ^5^ NBS target condition since October 2016 (N = 910,717). ^6^ One child with dual diagnosis, hypothyroidism and HPA, counted as a case for both conditions. ^7^ One child with dual diagnosis, CAH and MCADD, counted as a case for both conditions. Abbreviations: CACT, carnitine-acylcarnitine translocase deficiency; CAH, congenital adrenal hyperplasia; CF, cystic fibrosis (mucoviscidosis); CPT I/II, carnitine palmitoyl transferase I/II deficiency; GA I, glutaric acidemia type I; HPA, hyperphenylalaninemia; IVA, isovaleric acidemia; LCHADD, long-chain 3-hydroxyacyl-CoA dehydrogenase deficiency; MCADD, medium-chain acyl-CoA dehydrogenase deficiency; MSUD, maple syrup urine disease; NBS, newborn screening; PKU, phenylketonuria; PPV, positive predictive value; SCD, sickle cell disease; SCID, severe combined immunodeficiency; SMA, 5q spinal muscular atrophy; VLCADD, very-long-chain acyl-CoA dehydrogenase deficiency.

**Table 3 IJNS-11-00114-t003:** Age at initiation of care ^1^.

Condition	≤7 Days		8–10 Days		11–14 Days		>14 Days		Total
	n	%	n	%	n	%	n	%	n
Hypothyroidism	644	78.44	90	10.96	37	4.51	50	6.09	821
CAH	146	70.53	29	14.01	14	6.76	18	8.70	207
PKU	176	71.26	53	21.46	8	3.24	10	4.05	247
HPA	77	25.41	63	20.79	53	17.49	110	36.30	303
MCADD	150	53.20	62	21.80	29	10.20	42	14.84	283
Further IEMs, SCID, SMA	148	61.41	41	17.01	16	6.64	36	14.94	241
Total	1341	63.80	338	16.08	157	7.47	266	12.65	2102

^1^ Excluding CF and SCD. Abbreviations: CAH, congenital adrenal hyperplasia; CF, cystic fibrosis (mucoviscidosis); HPA, hyperphenylalaninemia; IEMs, inborn errors of metabolism; MCADD, medium-chain acyl-CoA dehydrogenase deficiency; PKU, phenylketonuria; SCD, sickle cell disease; SCID, severe combined immunodeficiency; SMA, 5q spinal muscular atrophy.

**Table 4 IJNS-11-00114-t004:** Participants of the Bavarian NBS long-term follow-up study (birth cohort 1999–2013). Further details are provided in Appendix A.

Condition	Individuals (n)
Hypothyroidism	385
CAH ^1^	106
Biotinidase deficiency	20
Classic galactosemia	20
PKU	144
MSUD	9
MCADD	164
LCHADD	5
VLCADD	15
Carnitine cycle defects	6
GA I	16
IVA	13
Total	903

^1^ One child with dual diagnosis of CAH and MCADD. Abbreviations: CAH, congenital adrenal hyperplasia; GA I, glutaric acidemia type I; IVA, isovaleric acidemia; LCHADD, long-chain 3-hydroxyacyl-CoA dehydrogenase deficiency; MCADD, medium-chain acyl-CoA dehydrogenase deficiency; MSUD, maple syrup urine disease; PKU, phenylketonuria; VLCADD, very-long-chain acyl-CoA dehydrogenase deficiency.

**Table 5 IJNS-11-00114-t005:** Metabolic or electrolyte decompensations in participants of the Bavarian NBS long-term follow-up study with conditions predisposing to decompensation (birth cohort 1999–2013). Further details on fatal decompensations are provided in Appendix A.

Condition	Individuals	Individuals with Metabolic or Electrolyte Decompensation		
	n	n	Including	
			Neonatal (n)	Fatal (n)
CAH	105	52	28	
CAH and MCADD	1	1		1
Biotinidase deficiency	20			
Classic galactosemia	20	9	9	
MSUD	9	4	1	
MCADD	164	11	3	2
LCHADD	5	5		2
VLCADD	15	2	2	
Carnitine cycle defects	6	4	2	2
GA I	16	3		
IVA	13	4		
Total	374	95	45	7

Abbreviations: CAH, congenital adrenal hyperplasia; GA I, glutaric acidemia type I; IVA, isovaleric acidemia; LCHADD, long-chain 3-hydroxyacyl-CoA dehydrogenase deficiency; MCADD, medium-chain acyl-CoA dehydrogenase deficiency; MSUD, maple syrup urine disease; VLCADD, very-long-chain acyl-CoA dehydrogenase deficiency.

**Table 6 IJNS-11-00114-t006:** Developmental benchmarks at age 6 [37] achieved by participants of the Bavarian NBS long-term follow-up study (birth cohort 1999–2013; N = 752).

Condition	Individuals	Developmental Benchmarks Achieved at Age 6		
	n	n	%	CI 95%
Hypothyroidism	299	276	92.3	89.3–95.3
CAH	99	95	96.0	92.1–99.8
PKU	120	109	90.8	85.7–96.0
MCADD	149	138	92.6	88.4–96.8
Further IEMs	85	62	72.9	63.5–82.4
Total	752	680	90.4	88.3–92.5
Reference population [37]	1755	2002	87.7	86.2–89.1

Abbreviations: CAH, congenital adrenal hyperplasia; CI, confidence interval; IEMs, inborn errors of metabolism; MCADD, Medium-Chain Acyl-CoA Dehydrogenase Deficiency; PKU, phenylketonuria.

**Table 7 IJNS-11-00114-t007:** Overview of main findings from the Bavarian NBS long-term follow-up study (birth cohort 1999–2013), stratified by the most common diagnoses.

Outcome Category	Hypo-thyr.	CAH	PKU	MCADD	Further IEMs	Total	Figure/Table
Growth	✔	↓	(↓)	✔	(↓)	✔	Figure 4, Appendix A
BMI	✔	↑	(↓)	✔	(↓)	(↑)	Figure 5, Appendix A
Development	✔	✔	✔	✔	↓	✔	Table 6, Appendix A Appendix A
Behaviour	✔	✔	✔	✔	✔	✔	Appendix A
Quality of Life	✔	✔	✔	✔	✔	✔	

Symbols and Abbreviations: ✔ finding comparable to reference population. ↓/↑ More frequently decreased or impaired/increased when compared to reference population. (↓)/(↑) Unclear trend: more frequently decreased/increased when compared to reference population. BMI, body mass index; CAH, congenital adrenal hyperplasia; IEM, inborn error of metabolism; Hypothyr., hypothyroidism; MCADD, medium-chain acyl-CoA dehydrogenase deficiency; PKU, phenylketonuria.

## Data Availability

The data presented in this study are available on request from the corresponding author. The data are not publicly available due to privacy restrictions.

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
