# Peer review of "A 25-Year Retrospective on Bavaria’s Newborn Screening Programme: Achievements, Challenges and Long-Term Follow-Up"

_2409-515X, 2025, doi:10.3390/ijns11040114_

Round 1
Reviewer 1 Report
Comments and Suggestions for Authors
“A 25-Year Retrospective on Bavaria’s Newborn Screening Programme: Achievements and Challenges” (IJNS-3904159)
General Summary
This manuscript presents a comprehensive 25-year retrospective analysis (1999–2023) of Bavaria’s newborn screening (NBS) programme and a longitudinal follow-up study covering the 1999–2013 birth cohorts. Drawing on data from over 2.8 million screened newborns, the authors provide an in-depth evaluation of screening performance, including coverage, timeliness, diagnostic accuracy, and long-term outcomes. The paper is clearly structured, methodologically rigorous, and addresses a highly relevant topic in public health and preventive medicine. It also highlights the distinctive organizational model of the Bavarian NBS system, characterized by centralized coordination, systematic tracking, and integrated follow-up.
Major Strengths
- Exceptional data completeness and scale – The dataset of 2.85 million newborns over 25 years is uniquely comprehensive, enabling robust evaluation of screening indicators such as recall rate, PPV, and condition-specific prevalence.
- High screening coverage and efficiency – The coverage rate of 99.83% and the 99.09% completion rate of requested repeat tests demonstrate outstanding programme performance and systematic tracking efficiency.
- Strong methodological design – The study combines population-based surveillance with long-term clinical follow-up, supported by well-defined quality assurance parameters and consistent analytical protocols.
- Integration of follow-up and outcome data – The inclusion of longitudinal follow-up results (up to 18 years) adds significant value, providing rare insight into the long-term medical, developmental, and psychosocial outcomes of NBS-identified children.
- Clear presentation and transparency – Tables and figures (e.g., recall rates, PPV stratification, developmental outcomes) are well organized and facilitate interpretation. The supplementary material further strengthens reproducibility.
- Public health relevance – The work convincingly illustrates how centralized coordination and close collaboration among laboratories, clinical centers, and public health authorities can achieve near-universal coverage and sustained follow-up.
Weaknesses / Points for Consideration
- Limited discussion of false negatives and residual risk – Although false-negative cases are described in detail, the discussion could further elaborate on how these findings inform current quality assurance and algorithm refinement.
- Generalizability – Given Bavaria’s unique institutional setting, the applicability of this organizational model to other regions could be discussed more explicitly.
- Terminology correction – In the Introduction, the phrase “dried blood spots (NBS)” should be corrected to “dried blood spots (DBS).”
Minor Suggestions
- Clarify whether the “recall rate” denominator refers to all screened newborns or to those with complete data entries (lines 213–214).
- Consider summarizing the main long-term outcome indicators (developmental, behavioural, quality of life) in a concise table in the main text, referring to Supplementary Table 4 and Figures 1–4.
- A short reflection in the Discussion on the evolving screening panel (e.g., addition of SCID, SMA,) and implications for programme adaptability would be useful.
Overall Assessment
This is an excellent and comprehensive manuscript that makes a valuable contribution to the field of newborn screening. It demonstrates the effectiveness of an integrated, population-based screening system with near-complete coverage and high-quality long-term follow-up. The study is well executed, the data are robust, and the conclusions are well supported. After minor textual and editorial corrections, I strongly recommend the paper for publication.
Author Response
We would like to express our gratitude to Reviewer 1 for their positive review of our manuscript and very helpful constructive suggestions. It has been endeavoured to consider all suggestions and implement them accordingly. The language has already been professionally reviewed and will be checked again in the final version. The figures have been adjusted for better readability and clarity. In the revised manuscript, additions are highlighted in blue font, and sections that have only been restructured are shaded in grey for ease of reference.
We thank the Reviewer again for their positive review of our manuscript and valuable comments that have helped improve our manuscript.
Please find a point-by-point response to the comments in the attached draft.

Reviewer 2 Report
Comments and Suggestions for Authors
The article presents 25 years of NBS results from the Bavarian Health and Food Safety Authority’s perspective. It reports an overall PPV exceeding 70%, with approximately 80% of affected infants receiving treatment within 10 days of birth. In addition, the study includes long-term outcomes from a birth cohort spanning 1999-2013. The topic is valuable and contributes to understanding the evolution and impact of regional NBS implementation. However, several areas require clarification and additional data presentation to enhance the manuscript’s transparency and reproducibility.
Major Comments
Section 2, Framework (Lines 76–185):
The framework section would benefit from a more structured presentation. A visual illustration (e.g., a flowchart or schematic) should be added, and only the key components need to be described in text. A comparison between the Bavaria program and the broader German NBS framework would also help readers understand regional differences.
Role of the Newborn Screening Center: It appears that the “newborn screening center” primarily functions in a coordinating capacity. If this is the case, its specific role should be emphasized and described in the Methods section, including how coordination and communication processes contribute to result validation and reproducibility.
Coverage Rate: One of the major achievements of this program is the high screening coverage. Please describe how coverage evolved over time and discuss the factors that contributed to its improvement.
Methods
Detailed longitudinal metrics are lacking except for recall rate. Please include data or trends over time, particularly for other key performance indicators.
The specific assays and cutoff values for each screened condition are not mentioned. These are essential for reproducibility and comparison across programs.
As the study is part of the broader German NBS framework, it would be useful to list relevant national data or references for context.
Long-term outcome data were primarily obtained from parental questionnaires. Please clarify how these cases are linked to the follow-up medical care and data validation through “medical experts.” This clarification is important for assessing data quality, particularly for parameters such as metabolic decompensation.
Results
Figure 1 , Recall Rate: Please indicate in the figure or its legend the years when two-tier CAH testing and other additional NBS programs were introduced. Also, explain the gradual decrease in CAH recall rates from 2008 to 2016 and the sharp reduction observed in 2017.
Table 3: The table should indicate how many of the initial 1,200+ newborns were excluded under each condition (e.g., transient hypothyroidism, mild HPA). Clarify whether this is a prospective or retrospective study.
Table 4, Metabolic Decompensation (Line 484): The numbers of both decompensation events and affected individuals are very low in this cohort. Adding data on deceased cases for each condition would enhance the clinical relevance. If possible, include comparative data from other NBS programs, since such comparisons are discussed later. Additional clinical case information would also help illustrate the value of NBS and this program’s contribution.
Underweight and Obesity Comparison (Lines 501-502): Please describe the statistical methods used for comparison with the reference population and include corresponding p-values.
Author Response
We would like to thank Reviewer 2 for their detailed review and valuable comments. All comments and suggestions have been given due consideration, and the manuscript has undergone several changes. The language has previously been professionally reviewed and will be subject to further review in the final version. The figures have been adjusted for better readability and clarity. In the revised manuscript, additions are highlighted in blue font, and sections that have been restructured are shaded grey for ease of reference.
The point-by-point response to the comments and suggestions is provided in the attachment, with changes made to the manuscript marked in red.

Reviewer 3 Report
Comments and Suggestions for Authors
This study reports 25 years of Bavaria’s newborn screening program, covering over 2.85 million infants with a 99.8% participation rate. The authors described that early diagnosis enabled timely treatment in about 2,500 cases, and systematic follow-up ensured long-term care for nearly all identified children. The results demonstrate that centralized coordination and collaboration can achieve near-universal coverage and sustained positive outcomes.
Major points
The manuscript provides a valuable overview of Bavaria’s NBS achievements and long-term outcomes, but its interpretive depth could be strengthened. Clarifying methodological points, addressing inconsistencies in data presentation, and expanding on time-dependent analyses would greatly improve the paper’s impact. Its novelty is limited and there is little analysis showing trends or improvements over time. Apart from Figure 1, few data demonstrate longitudinal progress or programmatic evolution. In Figure 5, parental assessments are presented by child age; however, parental perceptions may differ significantly between 1999 and 2013 due to advances in diagnosis and treatment. The authors should consider this temporal bias.
Minor points
Line 237
The authors mention long-term follow-up up to 18 years of age. Given that some cases are presumably still under follow-up, what was the rationale for publishing the study now? Do the authors expect that the main outcomes will remain unchanged even if follow-up continues until 2031?
Line 290
The statement that “approximately 75 test cards were lost annually in transit between the sender and the laboratory” requires clarification. Does this refer to lost dried blood spot samples, accompanying patient information, or both? Losing 75 samples per year seems non-trivial—what corrective actions have been implemented?
Line 303
The proportion of samples taking more than three days to reach the laboratory increased from 8.9% in 1999 to 23.2% in 2023. What factors contributed to this deterioration in transport efficiency, and what measures are planned to address it?
Line 398
For the cases in which samples were lost during transport, at what postnatal age was clinical intervention initiated? Were any preventive measures established to avoid similar transport losses in the future?
Line 475
The authors state that metabolic decompensation episodes occurred regardless of specialist follow-up. Was the severity or frequency of these episodes affected by whether or not the child was under specialist care? Furthermore, in cases not under specialist care, who made that decision—the parents or the attending physician?
Author Response
We would like to thank Reviewer 3 for their detailed review and valuable comments. All comments and suggestions have been given due consideration, and the manuscript has undergone several changes. The language has previously been professionally reviewed and will be subject to further review in the final version. The figures have been adjusted for better readability and clarity. In the revised manuscript, additions are highlighted in blue font, and sections that have been restructured are shaded grey for ease of reference.
The point-by-point response to the comments and suggestions is provided in the attachment, with changes made to the manuscript marked in red.
